# Trophoblast Cell Surface Antigen 2 (Trop2) Is Expressed in Cases of EBV-Positive Diffuse Large B-Cell Lymphoma Emerging from Angioimmunoblastic T-Cell Lymphoma

**DOI:** 10.3390/reports7020037

**Published:** 2024-05-14

**Authors:** Susanne Ghandili, Judith Dierlamm, Carsten Bokemeyer, Clara Marie von Bargen, Anne Menz, Sören Alexander Weidemann

**Affiliations:** 1Department of Oncology, Hematology and Bone Marrow Transplantation with Section Pneumology, University Cancer Center Hamburg, University Medical Center Hamburg-Eppendorf, Martinistrasse 52, 20246 Hamburg, Germany; dierlamm@uke.de (J.D.); bokemeyer@uke.de (C.B.); 2Institute of Pathology, University Medical Center Hamburg-Eppendorf, Martinistrasse 52, 20246 Hamburg, Germany; c.von-bargen@uke.de (C.M.v.B.); a.menz@uke.de (A.M.); s.weidemann@uke.de (S.A.W.)

**Keywords:** diffuse large B-cell lymphoma, trophoblast cell surface antigen 2, immunohistochemistry, tissue microarray, molecular landscape

## Abstract

Although trophoblast cell surface antigen 2 (Trop2)-targeting drugs are already approved or under investigation in various solid tumors, the significance of Trop2 in lymphoma is unknown. Thus, our objective was to investigate the expression of Trop2 in diffuse large B-cell lymphoma (DLBCL) through a systemic immunohistochemistry screening. We constructed a tissue microarray comprising tissue from 92 DLBCL patients, each diagnosed at the University Medical Center Hamburg-Eppendorf (2020–2022). Trop2-immunohistochemistry was carried out, and positive staining was deemed a specific membranous positivity. Four samples were derived from Epstein-Barr virus (EBV)-positive DLBCL, with one case of EBV-positive DLBCL following angioimmunoblastic T-cell lymphoma (AITL). Strong Trop2 immunostaining was detectable in 1 of 91 analyzable samples, originating from a patient with a composite EBV-positive DLBCL emerging from AITL. Therefore, we performed an additional database search to identify all cases of composite EBV-positive DLBCL emerging from AITL since 2015. Five additional cases were identified and stained for Trop2, revealing two cases with strong B-blast positivity. Our preliminary data imply that Trop2 appears absent in de novo DLBCL, whereas Trop2 is strongly expressed in cases of a rare variant of EBV-positive DLBCL. Further investigations are needed to confirm our results, particularly on the subset of EBV-positive DLBCL emerging from AITL.

## 1. Introduction

Non-Hodgkin lymphoma comprises a heterogeneous group of nodal, extranodal, or disseminated B- or T-cell malignancies; worldwide diffuse large B-cell lymphoma (DLBCL) accounts for approximately every third to fourth non-Hodgkin lymphoma [1,2,3,4]. Even though most DLBCL can be classified as primary DLBCL based on a de novo origin, a significant percentage of DLBCL originates or emerges from indolent B-cell lymphoma. Progression rates are in the range 2–8% in chronic lymphatic leukemia up to 14.3% in follicular lymphoma [5,6]. Moreover, DLBCL can originate within nodular lymphocyte predominant B-cell lymphoma or peripheral T-cell lymphoma (e.g., angioimmunoblastic T-cell lymphoma (AITL) [7,8,9,10,11,12]. Current treatment regimens that e.g., Pola-R-CHP or R-CHOP (consisting of polatuzumab vedotin, rituximab, cyclophosphamide, doxorubicin, and prednisolone and rituximab, cyclophosphamide, doxorubicin, vincristine, and prednisolone, respectively), have shown curability rates of up to 60–70% in patients with previously untreated DLBCL [3,13]. Yet, for chemotherapy-insensitive primary refractory or early relapsing DLBCL, promising newly developed treatment approaches, including CD19-directed CAR-T-cell therapies and T-cell-engaging bi-specific CD3xCD20-directed antibodies, have recently been approved, changing the treatment landscape of refractory and relapsing DLBCL [14,15,16,17,18,19]. However, a precision medicine strategy based on identifying potentially targetable driver mutations might provide additional treatment options in a subset of patients.

The tumor-associated calcium signal transducer 2, also referred to as trophoblast cell surface antigen 2 (Trop2), is a cell-surface glycoprotein, first elucidated as a transmembrane calcium signal transducer, encoded by the TACSTD2 gene [20]. Trop2 expression was initially delineated in non-cancerous trophoblasts and fetal tissues and is physiologically expressed in the stratified squamous epithelium of the skin, tonsillar crypts, esophagus, and uterine cervix [20,21,22]. Overexpression of Trop2 has been reported in glioma, basal cell carcinoma, squamous cell cancer of the skin, squamous cell cancer of the oral cavity, head, and neck, thyroid, lung, esophageal, gastric, colorectal, pancreatic, renal, breast, uterine, and ovarian cancer [23,24]. Moreover, in several solid malignancies such as pancreatic, hilar cholangiocarcinoma, cervical and gastric cancer, high Trop2 expression correlates with poor prognosis [19]. Furthermore, Trop2 interacts with several molecular signaling pathways associated with tumor development and progression (e.g., MAPK, PI3K/AKT). Levels of Trop2-mRNA expression correlate with tumor size and, in some cancer entities, with the extent of metastasis, whereas Trop2 knockdown or antibody-based inhibition leads to decreasing in vitro cancer cell proliferation, invasion, colony formation, and migration [24]. Since Trop2 is a transmembrane protein with an extra-cellular domain overexpressed in various solid malignancies compared to normal tissues, it has been the subject of targeted therapy. Sacituzumab govitecan is a first-in-class Trop2-directed antibody-drug-conjugate delivering SN38 (an irinotecan metabolite), first approved in 2021 “for the treatment of patients with unresectable locally advanced or metastatic triple-negative breast cancer who have received two or more prior systemic therapies, at least one of them for metastatic disease”, shortly after “for locally advanced or metastatic urothelial cancer who have previously received platinum-containing chemotherapy and either programmed death receptor-1 or programmed death-ligand 1 inhibitor”, and most recently “for patients with unresectable locally advanced or metastatic hormone receptor-positive, HER2-negative breast cancer who have received endocrine-based therapy and at least two additional systemic therapies in the metastatic setting” [24,25,26,27]. Another promising Trop2-directed antibody-drug conjugated is datopotamab deruxtecan, which is currently part of clinical trials (e.g., NCT04526691, NCT05555732, NCT05489211, and NCT05374512). Even though Trop2 has been widely investigated in preclinical and clinical trials in the setting of solid tumors, studies on its role in hematological malignancies, particularly lymphoma, remain rare [28,29,30]. Thus, up to now, Trop2 expression in lymphoma and DLBCL has yet to be investigated. 

Since sacituzumab govitecan, an approved Trop2-targeting drug, is now available, we sought to examine the expression of Trop2 by performing a systemic immunohistochemistry (IHC) screening on a tissue microarray (TMA) derived from 92 patients with newly diagnosed or refractory or relapsed (r/r) DLBCL.

## 2. Materials and Methods

### 2.1. Patients

In this observational, single-center study, we examined 92 consecutive DLBCL patients aged 18 years or older who were immunohistochemically tested for Trop2 overexpression. This analysis adds to our previously published research, in which we performed a systematic immunohistochemical screening for NTRK fusion proteins on the same DLBCL cohort and accordingly using the same DLBCL-TMA. Details regarding the patient cohort can be found in Ghandili et al. [2]. As previously described, samples of all patients included in this analysis were derived from the Institute of Pathology of the University Medical Center Hamburg-Eppendorf, Germany, during the years 2020, 2021, and 2022. All patients were diagnosed with DLBCL as part of the clinical routine. The study design is presented in Figure 1. For disease staging assessment, we used the Ann Arbor classification [31]. For the calculation of prognostic risk scores, we used the International Prognostic Index (IPI) [32]. 

The date 21 December 2022 marked the data cutoff. Five additional cases date from 2017–2022. Representative tumor material was stained on large-area sections using the identical protocol, as explained for the TMA block below.

The use of archived remnants of diagnostic tissues for manufacturing of tissue microarrays (TMA) and their analysis for research purposes, as well as patient data collection and analysis, were reviewed and approved according to local legal requirements (HmbKHG, §12) and by the local ethics committee (Ethics Commission Hamburg, WF-049/09). Moreover, all analyses were carried out in compliance with the Declaration of Helsinki.

### 2.2. Tissue Microarray (TMA)

An already existing TMA of 92 DLBCLs was utilized. For the years 2020–2022, the institutional pathology database included a total of 414 samples of diagnosed DLBCL. Thereof, 92 samples were suitable for the TMA construction. Tissue samples of deficient sample size were excluded. The leading causes for deficient tissue sample size were either entirely or nearly completely utilized tissue during diagnostic workup or samples gained by small needle biopsy containing too little material for tissue extraction. Furthermore, we excluded all bone marrow biopsies since they are generally unsuitable for TMA construction.

TMA construction has already been discussed in detail [33]. Briefly, tissue cylinders, each 0.6 mm in diameter, were taken from tumor-containing areas of “donor” tissue blocks and placed into empty recipient paraffin blocks. The squamous epithelium of the skin served as a positive control.

### 2.3. Immunohistochemistry (IHC)

A freshly cut slide of the TMA block was deparaffinized with xylol and rehydrated through an alcohol series. A heat-induced antigen retrieval for 5 min was performed in an autoclave at 121 °C in pH 7.8 using DakoTarget Retrieval SolutionTM (Agilent, Santa Clara, CA, USA; #S2367). Endogenous peroxidase activity was blocked with Dako Peroxidase-Blocking SolutionTM (Agilent, Santa Clara, CA, USA; #52023) for 10 min. A primary antibody specific against TROP2 protein (recombinant rabbit monoclonal, MSVA-733R, MS Validated Antibodies, Hamburg, Germany) was applied at 37 °C for 60 min with a dilution of 1:150. According to the manufacturer’s protocol, the bound antibody was visualized using the EnVision KitTM (Agilent, Santa Clara, CA, USA; #K5007). The slide was then counterstained with haemalaun. The percentage of Trop2-positive lymphoma cells for each spot was estimated, and the staining intensity was semiquantitatively recorded (no staining = 0, weak staining = 1+, moderate staining = 2+, strong staining = 3+). 

### 2.4. Endpoints

The primary objective of this study was to assess the frequency and intensity of immunohistochemical Trop2 expression in tumor samples of DLBCL patients. 

### 2.5. Statistical Analysis

The statistical analyses were conducted using Microsoft Excel for Mac, version 16.38 (Microsoft Cooperation, Redmon, WA, USA). Continuous values are expressed as median, and nominal variables are presented as numbers. 

## 3. Results

### 3.1. Patient Characteristics

Details of patient characteristics have been previously reported. In summary, all samples of 91 consecutive patients diagnosed with DLBCL were included in the analysis. Patient demographics and characteristics are presented in Table 1. Most samples originated from lymph nodes (38%), followed by testes (9%). The remaining samples were derived from other extranodal organs, such as the small intestine, colon, skin, spleen, liver, nasal, and buccal mucosa. The median age was 74, ranging from 32 to 90, and predominantly male (60%). In 13 patients, DLBCL derived or emerged from another preexisting lymphoma, including follicular lymphoma, lymphoplasmacytic lymphoma, chronic lymphatic lymphoma, and one case of AITL. 

### 3.2. Immunohistochemical Findings

Based on identifying one case of strongly stainable Trop2-positive EBV-positive DLBCL (Figure 2 and Figure 3) that emerged from AITL, next, we performed an additional systemic search in the pathology reports database to identify all cases of EBV-positive DLBCL that emerged from AITL. A total of five other patients were identified. Characteristics and IHC results of these additional five patients are presented in Table 2. Two cases had strong membranous staining on the B-blasts, while all T-cells remained negative (Figure 4).

## 4. Discussion

Up to now, the role of Trop2 in lymphoma, especially in DLBCL, has been sparsely examined. Thus, our objective was to investigate the frequency and intensity of Trop2 expression in DLBCL by performing a systemic immunohistochemical screening for Trop2 expression in a sizeable cohort of DLBCL samples. Trop2 expression was detectable in only 1 of 91 TMA samples. Remarkably, the positive sample originated from a patient with secondary EBV-positive DLBCL developed in the background of AITL. By supplementary investigation of five additional samples of EBV-positive DLBCL emerging from AITL available at our department, we could observe strong Trop2 expression in two of these cases. 

Based on a suspected dysfunction of both B- and T-cells, AITL is characterized by the development of concomitant autoimmune disorders and a particular tendency of patients to develop opportunistic infections. Thus, the systemic immunological dysfunction observed in AITL is reminiscent of the late stages of infection by the human immunodeficiency virus [11]. Furthermore, chemotherapy protocols impair the immune system and inhibit cellular immunity, favoring the growth of latent EBV-infected B cells. EBV-positive immunoblasts are a common diagnostic finding in AITL. Thirty percent of AITL cases develop clonal IgG rearrangements, and up to 5% of AITL cases have a morphologically perceptible EBV-positive B-cell lymphoproliferation. Nevertheless, transformation to a high-grade lymphoma, e.g., secondary DLBCL, is possible but rarely occurs [12,34]. Unsurprisingly, in immunohistochemistry, only a few blasts seemed to stain positive. In all positive cases, several follow-up bone marrow and lymph node biopsies were taken within a few months. First, a clonal B-cell lymphoproliferation of unclear significance was detected, which was considered to progress to a DLBCL after persisting in follow-up biopsies. Naturally, in such cases there were no dense, homogenous areas of blasts, but rather small clusters of blasts. Therefore, the expected number of positive B-blasts was lower than in de novo DLBCL.

The fact that an immunosuppressive environment contributes to the clonal proliferation of EBV-positive cells from which high-grade lymphomas can arise is already well described in the literature for the post-transplant scenario [35,36,37]. The current edition of the WHO has dedicated a separate chapter to this phenomenon and replaced the former term post-transplant lymphoproliferations (PTLD) with the broader term lymphoproliferations and lymphomas associated with immunodeficiency and dysregulation (IDD lymphomas). We consider it appropriate to classify the cases of EBV-positive secondary DLBCL described here in this category and to assume a similar pathogenesis of insufficient immune regulation within the tumor environment of AITL.

Regarding de novo EBV negative DLBCL, our results generally align with those previously published by Dum and colleagues [28]. Dum et al. report one of the few studies of immunohistochemically screening for Trop2, among others, in a large cohort of different lymphoma entities, including 108 samples of DLBCL. Interestingly, Trop2 was neither detectable in DLBCL nor in any other lymphoma entity, including Hodgkin, indolent B-cell, mantle cell, and T-cell lymphoma [28]. Another analysis investigating the expression of Trop2 in lymphoma is a Chinese study reported by Chen et al. The authors demonstrated a high immunohistochemical Trop2 expression in 14 of 26 tissue samples of extranodal NK/T cell lymphoma, nasal-type patients [30].

Albeit we performed a systemic immunohistology screening for Trop2, our study has limitations, since there remains a residual risk of a potential selection bias of the used DLBCL samples gained from a single center cohort. In addition, data on patients’ clinical characteristics were only available for a subset of patients since even though all patients were diagnosed, they were not compulsively treated at our university medical center.

Since we observed no space-filling but rather sporadic Trop-2 positive blast population, predictions regarding potentially Trop2-directed treatment response remain content of further research.

## 5. Conclusions

In conclusion, even though Trop2 was not detectable in EBV-negative de novo DLBCL, we were able to immunohistochemically detect high Trop2 expression in a total of three samples of EBV-positive DLBCL transformed from AITL. To our knowledge, this is the first analysis reporting high Trop2 expression in secondary EBV-positive DLBCL. Nevertheless, additional studies of Trop2 in cohorts of EBV-positive DLBCL, particularly those emerging from AITL of similar or larger size, are necessary to confirm our results.

## Figures and Tables

**Figure 1 reports-07-00037-f001:**
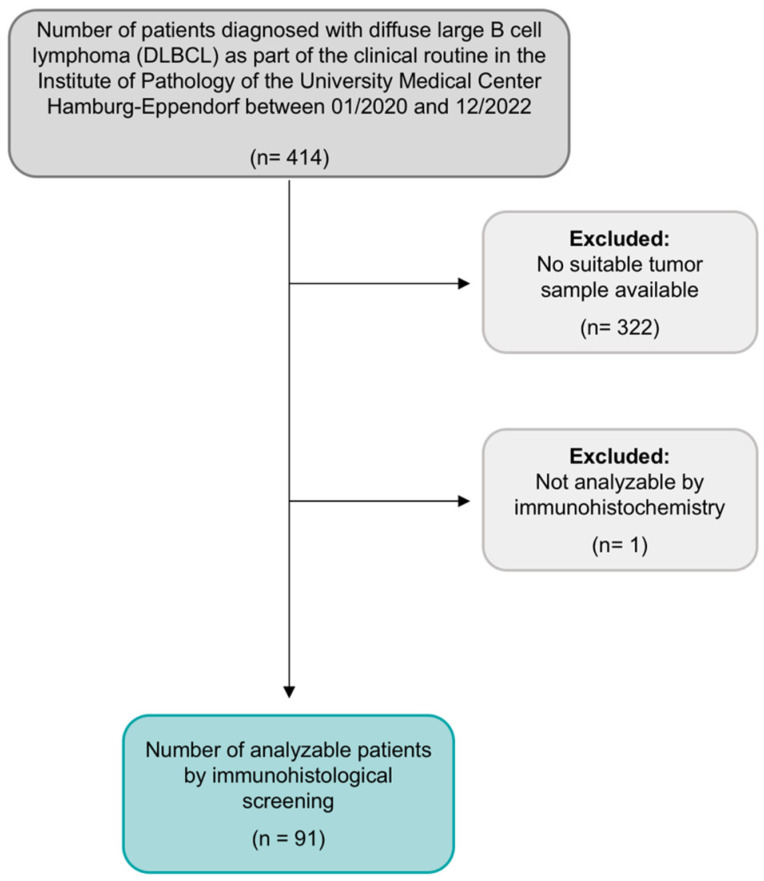
Flow chart of study design and population, as previously published by Ghandili et al. 2023 [2].

**Figure 2 reports-07-00037-f002:**
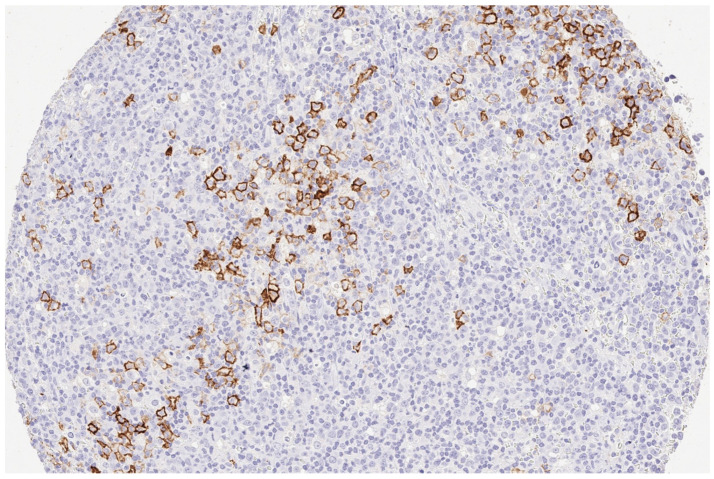
(200×): Strong (3+) immunohistology Trop2 expression in a case of composite EBV-positive DLBCL emerging from AITL.

**Figure 3 reports-07-00037-f003:**
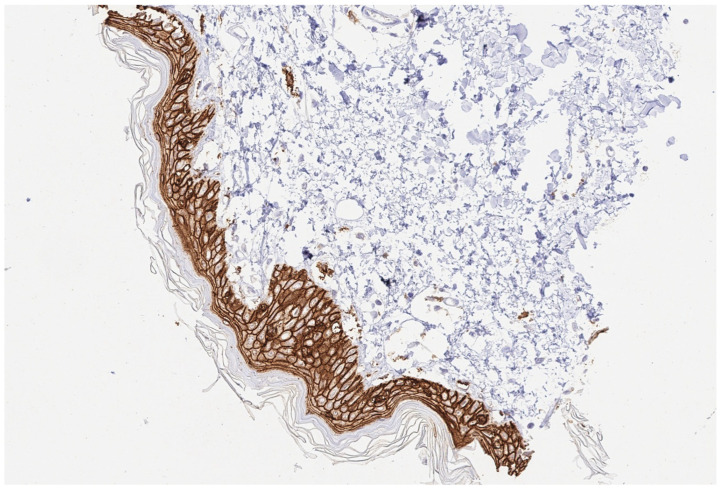
(200×): Trop2 positive control tested on a skin sample.

**Figure 4 reports-07-00037-f004:**
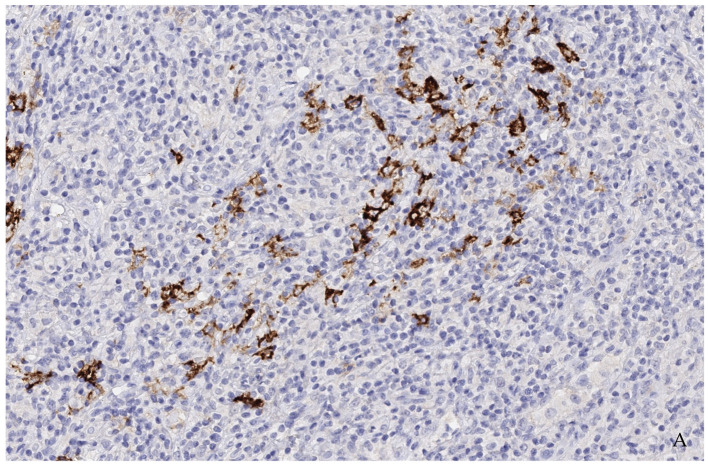
(**A**,**B**) (200×): Case 1 (**A**) and Case 4 (**B**): Most cells are T-cells of angioimmunoblastic T-cell lymphoma as well as reactive background cells. Single cells and incipient groups of B-blasts were consistently found to moderately to strongly express Trop2. In additional studies, these blasts showed strong positivity for CD30 and EBV-encoding RNA (EBER) (not shown) as well as a clonal IgH-genre arrangement and indicated a transition from a clonal B-lymphoproliferation into high-grade EBV-positive B-cell lymphoma.

**Table 1 reports-07-00037-t001:** Patient characteristics of the TMA cohort [1].

**Total number of analyzable patients, n**	91
**Age at DLBCL diagnosis, median (range)**	74 (32–90)
**Female sex, n (%)**	37 (40)
**Subtypes, n (%)**	
Cell of origin type: Germinal center B-cell-like	41 (45)
Epstein-Barr-virus positivity	4 (4)
Double hit	2 (2)
Triple hit	0
**Ann Arbor stage, n (%)**	
1A or 1B and 2A or 2B	18 (20)
3A or 3B	5 (5)
4A or 4B	14 (15)
Not evaluable	59 (65)
**International Prognostic Index, n (%)**	
0–1	12 (13)
2–3	13 (14)
4–5	4 (4)
Not evaluable	62 (68)

**Table 2 reports-07-00037-t002:** Characteristics and IHC results of additional five patients/tissue samples of composite EBV-positive DLBCL emerging AITL. All cases of composite EBV-positive DLBCL were confirmed by reference pathology.

	Age in Years	Sex	Tissue Origin	Trop2 Staining Intensity	Refer to Figure
Case 1	56	Female	Lymph node	3+	4A
Case 2	84	Female	Lymph node	0	NA
Case 3	48	Female	Lymph node	0	NA
Case 4	81	Female	Lymph node	3+	4B
Case 5	48	Female	Skin	0	NA

NA: Not applicable.

## Data Availability

The data generated in this study are available upon request from the corresponding author.

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
