# Peer review of "Trophoblast Cell Surface Antigen 2 (Trop2) Is Expressed in Cases of EBV-Positive Diffuse Large B-Cell Lymphoma Emerging from Angioimmunoblastic T-Cell Lymphoma"

_reports, 2024, doi:10.3390/reports7020037_

Round 1

Reviewer 1 Report

Comments and Suggestions for Authors

The authors discuss the extremely important problem of anticancer targeted therapy. New methods for identifying molecules key to carcinogenesis are constantly being discovered and improved like as TMA. 

However I have some doubts:

1. abstract: mistake identi-fied 

2. material and methods: "For each spot, the percentage of Trop2-positive lym- 138 phoma cells was estimated, and the staining intensity was semiquantitatively recorded 139 (no staining = 0, weak staining = 1+, moderate staining = 2+, strong staining = 3+"

how do you identify scale? you should add information about it.

3.  please add more specific information about  Ann Arbor stage, n (%) and IPI, n (%) in material and methods 

4. results : why don't you add results of  "IHC results of these additional five patients are presented in Table 2. T" of case 2,3 and 5 as a negative control?

Reviewer 2 Report

Comments and Suggestions for Authors

In this current article “Trophoblast cell surface antigen 2 (Trop2) is expressed in cases 2 of EBV-positive diffuse large B-cell lymphoma emerging from 3 angioimmunoblastic T-cell lymphoma”. The authors claim that Trop2 appears to be absent in de novo DLBCL, whereas Trop2 is strongly expressed in cases of a rare variant of EBV-positive DLBCL. The study looks interesting and in order to capture the audience interest few things can be added and thar article can be published.

1)Authors should introduce the topic to the readers firstly by mentioning about Non-Hodgkin Lymphoma and then mentioning DLBCL is a type of it.

2)Authors should mention the year with the cited author Ghandili et al., in the Fig. 1.

3)Is it the Declaration of Helsinki or Helsinki Declaration? Does it have any other more specification related to the year or Medical Association?

4) Which statistical method was used for analysing the quantitative score?

5)  Authors need to mention how they differentiate the staining intensity from a scale of 0 to 3+?

6) Is there any correlation between TROP2 and the prognostic markers of DLBCL? If yes, please do mention about that.

7) Quantitative expression of TROP2 would give a more clear picture of the expression studies.
